Association between toxic heavy metals and noncancerous thyroid disease: a scoping review

http://orcid.org/0009-0000-7888-9565 Rafi’i Muhammad Ridzwan p137727@siswa.ukm.edu.my
http://orcid.org/0000-0003-0007-6008 Ja’afar Mohd Hasni drmhasni@ukm.edu.my
http://orcid.org/0000-0002-0009-5244 Mohammed Nawi Azmawati
Md Hanif Shahrul Azhar
http://orcid.org/0000-0002-9537-8668 Md Asari Siti Najiha
Public Health Medicine Department, National University Malaysia , Cheras, Kuala Lumpur , Malaysia
Avino Pasquale
Electronic publication date: 2025 Feb 11
Publication date: 2025
Volume: 13
Electronic Location ID: e18962
Received 2024 Sep 5; Accepted 2025 Jan 20
Copyright: © 2025 Rafi’i et al.
Copyright year: 2025
Copyright holder: Rafi’i et al.
License: This is an open access article distributed under the terms of the Creative Commons Attribution License, which permits unrestricted use, distribution, reproduction and adaptation in any medium and for any purpose provided that it is properly attributed. For attribution, the original author(s), title, publication source (PeerJ) and either DOI or URL of the article must be cited.
License URL: https://creativecommons.org/licenses/by/4.0/

Keywords: Metal, Goiter, Thyroid nodule, Thyroid cyst, Hypothyroidism, Hyperthyroidism, Relationship, Endocrine, Pollution, Environment

Funding: The authors received no funding for this work.

==============================
Background

Toxic heavy metals such as chromium (Cr), arsenic (As), cadmium (Cd), mercury (Hg), and lead (Pb) are known to be priority pollutants due to their high degrees of toxicity and widespread presence in the environment. This review aimed to explore the association between heavy metals and noncancerous thyroid diseases by synthesizing findings from observational and experimental studies. This review addressed a critical intersection of environmental health, endocrinology, and public health. The findings would be of interest to a wide range of disciplines given the ubiquitous presence of toxic heavy metals in the environment and their potential to disrupt endocrine systems. The evidence-based information from diverse fields generated from this review will provide insights into the health implications of heavy metal exposure on thyroid function and guide the necessary interdisciplinary research and collaborative interventions.

Method

Three databases were searched, namely PubMed, Web of Science, and Scopus. The Arksey and O’Malley (2005) framework was used as a guide in conducting this scoping review. The reporting was carried out based on the Preferred Reporting Items for Systematic Reviews and the Meta-Analyses Extension for Scoping Reviews (PRISMA). The literature search retrieved 552 articles and 29 articles were included in the final review.

Results

As high as 83% of the 29 included studies followed an observational study design while the rest were experimental animal studies. Among the observational studies, two-thirds (66%) were cross-sectional studies while the rest were case-control studies (31%) and cohort studies (n = 1, 3%). Few number of studies in this review reported a significant association between Cr, As, Cd, Hg, and Pb with noncancerous thyroid diseases (2, 3, 16, 8, and 12) while another few (5, 8, 9, 5, and 11) did not show any significant association.

Conclusion

A heterogeneous and diverse sample population in the included studies could have potentially led to mixed findings about the association between toxic heavy metals and thyroid diseases in this review. Therefore, future research should prioritize longitudinal studies and controlled clinical trials to better elucidate the causative mechanisms and long-term impact of heavy metal exposure on thyroid health.

Introduction

Toxic heavy metals such as chromium (Cr), arsenic (As), cadmium (Cd), mercury (Hg), and lead (Pb), are known to be priority pollutants due to their high degree of toxicity and widespread presence in the environment (Balali-Mood et al., 2021; Unsal, 2018; Unsal et al., 2020; Zhang et al., 2024). They are considered systemic toxicants that can induce multiple organ damage even at low exposure levels, subsequently impacting critical physiological functions in the human body (Tchounwou et al., 2012). Their presence in water, soil, and air has rendered them significant environmental contaminants, with human exposure potentially occurring through ingestion, inhalation, or dermal contact. Among the many organs affected by heavy metal toxicity, the thyroid gland is particularly susceptible (Jaishankar et al., 2014). However, the specific effects of these metals on the thyroid gland, particularly in terms of noncancerous thyroid diseases, remain underexplored.

Noncancerous thyroid diseases, including benign thyroid nodules, goiter (thyroid gland enlargement), thyroiditis (thyroid gland inflammation), and thyroid cysts, are commonly perceived as less severe compared to thyroid cancer due to their lower mortality rates. Nevertheless, the global prevalence of noncancerous thyroid diseases can reach as high as 24%, hence making them a significant public health concern (Mu et al., 2022). Even with a low mortality rate, the morbidity of these conditions is still substantial. The thyroid gland is highly vulnerable to heavy metals due to its high vascularization, which leads to the active uptake of essential trace elements that share chemical similarities with toxic metals (Bano et al., 2019). These diseases often lead to chronic discomfort that requires lifelong medical management, such as regular thyroid level monitoring, ultrasound for thyroid nodules, and anti-inflammatory medications for thyroiditis (Ross et al., 2016). With alarming potential links between these diseases and exposure to environmental toxic heavy metals, there is a high burden on public health systems (Al-Bazi et al., 2021; Sun et al., 2019). With devastating impact on individual health and socioeconomic structures such as management for chronic conditions. Despite the non-lethal nature of these diseases, the chronicity of these conditions underscores the importance of understanding their potential risk factors, especially environmental exposures.

In the literature, recent studies have suggested that exposure to heavy metals may alter thyroid hormone synthesis, secretion, and metabolism, leading to conditions such as hypothyroidism, hyperthyroidism, and thyroid autoimmunity. The interplay between environmental factors and individual susceptibilities, such as genetic predisposition and nutritional status (e.g., iodine intake), also underscores the complexity of these associations (Ge et al., 2023; Wang et al., 2020). In a recent survey among U.S. adults, 7.2% of the participants exhibited abnormal thyroid function, with a significant portion being subclinical thyroid diseases. This study also highlighted the association between heavy metal exposure and thyroid dysfunction, particularly among populations exposed to Pb and Cd (Liao & Zhang, 2019). Similarly, another smaller-scale study focusing on miners in Ghana found that more than half (58.4%) of them showed blood Hg levels exceeding the occupational exposure threshold (≥5 μg/L), as well as a significant reduction in T3 and T4 levels among the exposed group (Afrifa, Ogbordjor & Duku-Takyi, 2018). Based on these two studies, it is vital to understand the interplay between heavy metals and thyroid function so that at-risk populations can be identified to implement targeted interventions.

To date, there have been numerous evaluations investigating the connection between ambient heavy metals and thyroid cancer (Gianì et al., 2019; van Gerwen et al., 2022; Vigneri, Malandrino & Vigneri, 2015; Webster et al., 2024; Zhang et al., 2019). However, none focused on the link with noncancerous thyroid diseases. Thus, this review aimed to explore the association between heavy metals and noncancerous thyroid diseases by synthesizing findings from observational and experimental studies. This review explores a key intersection between environmental health, endocrinology, and public health. Its findings will be relevant to various disciplines due to the widespread presence of toxic heavy metals in the environment and their potential to interfere with endocrine systems. By synthesizing evidence from multiple fields, the review aims to shed light on the health impacts of heavy metal exposure on thyroid function and highlight the need for interdisciplinary collaborative efforts in public health practice and future research.

Materials and Methods

The Arksey & O’Malley (2005) framework was used as a guidance for this scoping review and the reporting was conducted using the Preferred Reporting Items for Systematic Reviews and the Meta-Analyses Extension for Scoping Reviews (PRISMA) (O’Dea et al., 2021). Action listed in the particular (Arksey & O’Malley, 2005) framework was implemented.

Arksey & O’Malley (2005) framework

Defining the research question

This review explores the intricate relationship between exposure to environmental heavy metal pollution and noncancerous thyroid diseases, driven by the imperative to understand the impact of environmental factors on thyroid health. This review set out to determine the association between environmental heavy metals pollution (Cr, As, Cd, Hg, and Pb) and noncancerous thyroid disease. Environmental heavy metals with elevated density levels can easily pervade water, soil, and air matrices, hence making them ubiquitous contaminants in the environment. Human exposure to these heavy metals can occur through multiple routes, including ingestion, inhalation, or dermal contact, culminating in harmful effects on the thyroid. The term “noncancerous thyroid diseases” encapsulates a spectrum of benign pathological conditions such as thyroid nodules, goiter (thyroid gland enlargement), thyroiditis (thyroid gland inflammation), and thyroid cysts. Despite their benign nature, noncancerous thyroid disorders can manifest with debilitating symptoms, including respiratory compromise, neck discomfort, dysphagia, and hormonal dysregulation, thereby compromising the health and well-being of affected individuals. Furthermore, noncancerous thyroid diseases can have a significant and diverse impact on quality of life, affecting several facets of social, emotional, and physical health. Even though noncancerous thyroid diseases typically do not pose an immediate threat to health, meticulous long-term management tailored to the specific diagnosis is necessary.

Identifying relevant studies

Three databases were searched for this review, including PubMed, Web of Science, and Scopus. Table 1 shows the specific search string developed for this purpose. Initially, a total of 552 initial articles were gathered from the databases. We included all original research investigating the connection between environment heavy metals and noncancerous thyroid illness published between 2000 to 2024 (Song et al., 2023). However, books, book chapters, perspectives, letters to the editor, reviews, and other types of articles that lacked primary data or relevant outcomes were not excluded. Conference abstracts and proceedings were also disregarded because of the preliminary nature of the materials and the possibility of duplication with published articles. Non-English-language articles were also not included.

Table 1 Search string.

Database	Search string	
Scopus	TITLE-ABS-KEY ((“thyroid”) AND (“environment” OR “contamination” OR “pollution”) AND (“heavy metal”))	
Web of Science	(ALL = “thyroid gland” OR (ALL = thyroid AND ALL = gland) OR ALL = “thyroid gland” OR ALL = thyroid OR ALL = “thyroid usp” OR (ALL = thyroid AND ALL = usp) OR ALL = “thyroid usp” OR ALL = thyroids OR ALL = “thyroids” OR ALL = thyroidal OR ALL = thyroideal OR ALL = thyroidism OR ALL = thyroiditis OR ALL = thyroiditis OR ALL = thyroiditides) AND (ALL = environment OR ALL = contamination OR ALL = pollution) AND (ALL = heavy AND ALL = metal)	
PubMed	(“thyroid gland”[MeSH Terms] OR (“thyroid”[All Fields] AND “gland”[All Fields]) OR “thyroid gland”[All Fields] OR “thyroid”[All Fields] OR “thyroid usp”[MeSH Terms] OR (“thyroid”[All Fields] AND “usp”[All Fields]) OR “thyroid usp”[All Fields] OR “thyroids”[All Fields] OR “thyroid s”[All Fields] OR “thyroidal”[All Fields] OR “thyroideal”[All Fields] OR “thyroidism”[All Fields] OR “thyroiditis”[MeSH Terms] OR “thyroiditis”[All Fields] OR “thyroiditides”[All Fields]) AND (“environment”[All Fields] OR “contamination”[All Fields] OR “pollution”[All Fields]) AND (“heavy”[All Fields] AND “metal”[All Fields])	

Study selection

The retrieved studies were compiled using Endnote 20 (Clarivate). The search results from the three main electronic databases were downloaded. Initially, duplicate entries were found and removed with Endnote 20 (automation tools) before being checked manually by the main author to ensure all duplicate entries had been removed. M.R. and M.H. screened all the titles and abstracts to identify relevant research. Next, using the inclusion and exclusion criteria, the full texts of the selected publications were obtained and examined. The reference lists of all the included publications were examined to identify any research that might have been missed in the initial search. In addition to studies focusing on biological matrices such as blood and urine, we also included studies that analyzed thyroid tissue samples if they investigated histological or functional changes in the thyroid gland as a result of exposure to toxic heavy metals. However, studies that solely measured heavy metal concentrations in thyroid tissues without assessing their impact on thyroid structure or function were excluded.

Charting the data

Using a standardized table, M.R. and M.H. gathered the relevant information from the selected studies, including the names of the researchers, publication years, study designs, disease models used, heavy metal names, laboratory testing, heavy metals concentration, effect size of association between environmental heavy metals and noncancerous thyroid disease, biomarkers used (e.g., blood, urine, or thyroid tissue), as well as other important findings. For studies involving thyroid tissue, additional details on histological changes and tissue-level functional markers were also recorded to provide mechanistic insights.

Collating, summarizing, and reporting the results

The main objective of a scoping review is to summarize the developments in a specific field of study. Due to the wide variation in the included studies and the variables of interest in our review, a scoping review approach was utilized to summarize the main search results instead of synthesizing any specific elements. As a result, the major findings, namely the heavy metal types and levels, laboratory equipment used, as well as the association between the heavy metals with noncancerous thyroid disease in each study were outlined in the Results section below. Research gaps were also examined to identify the areas in need of further research and to generate recommended preventive actions.

Results

Three main databases were searched in this review (Web of Science, PubMed, and Scopus). The initial search found 552 articles (Web of Science: n = 197, PubMed: n = 130, Scopus: n = 225). Following the removal of 216 duplicates, 336 items underwent title and abstract screening. For a variety of reasons, 300 entries were excluded (irrelevant title = 131, not thyroid disease = 42, thyroid cancer = 21, review = 61, different population i.e., animals = 32, other heavy metals types n = 12). The remaining 36 articles underwent full-text screening, whereby three articles were excluded due to non-eligibility criteria (conference abstracts = 2, non-English publication = 3, proceeding = 2). In total, 29 studies were included in the final review. Figure 1 shows the article selection procedure.

Figure 1 Article selection.

Study characteristics

Among the 29 included studies (Table S1), the majority of the studies (83%) were observational studies while the remaining (17%) were experimental animal studies. Two-thirds of the observational studies were of cross-sectional design (66%), followed by case-control studies (31%) and one cohort study (3%). Biomarker selection is vital in determining the measurement of the environmental exposure of heavy metals in the human body. The majority of the studies used blood samples (45%) for heavy metals analysis while another 34% of the studies relied on urine samples for a similar purpose. Only 29% of the studies employed both urine and blood samples as biomarkers. The three main diagnostic methods for heavy metals level measurement in this review were inductively coupled mass spectrometry (ICP-MS), atomic absorption spectrophotometer, and inductively coupled plasma emission spectrometer (ICP-OES). As for the study participants, this review encompassed workers, children, pregnant mothers, and teenagers among the target population of interest.

Study outcomes

For thyroid disorders, a study found an association between thyroid cysts with Hg exposure. Adolescents having the highest tertile of Hg were 21.3 times more likely to have thyroid cysts compared to those in the first tertile (CI [2.2–207.0]) (Yalçin et al., 2022). Another study found that the level of Pb (mean 1.84 μg/dL, 95% CI [0.97–2.70]) was significantly lower in individuals with nodular goiter (p < 0.05) (Li et al., 2017). Besides that, the association between the effects of different toxic heavy metals and thyroid hormones was also examined in the remaining studies. The results are outlined in Table S2.

Chromium

Cr was examined in seven studies, of which two showed a significant association between Cr and thyroid hormone levels (Castiello et al., 2020; Nascimento et al., 2018). Another five studies did not report any significant association between Cr and thyroid hormone levels (Al-Bazi et al., 2021; Guo et al., 2018; Meeker et al., 2009; Sun et al., 2019; Xu et al., 2019). A study by Nascimento et al. (2018) found that blood levels of Cr (r = 0.476, p < 0.01) were positively correlated with thyroid stimulating hormone (TSH) concentrations and negatively associated (r = −0.333, p < 0.05) with free thyroxine (fT4) levels in the low exposure period. Another study by Castiello et al. (2020) found a significant association between the level of Cr with decreased TSH (β = −24, 95% CI [−42 to −1]).

Arsenic

Three studies in this review found a significant association between As and thyroid hormone concentrations in the human body. However, another eight studies did not detect any significant association between As and thyroid hormones (Campos et al., 2021; Castiello et al., 2020; Jurdziak et al., 2018; Liao & Zhang, 2019; Nascimento et al., 2018; Wang et al., 2020; Xu et al., 2019; Yalçin et al., 2022). A study by Meeker et al. (2009) found that As was associated with an increase in TSH (β = 0.25 (0.03, 0.47)). Another study by Guo et al. (2018) found that As was significantly linked to decreased levels of T3 (β = −0.95 (−3.70, 1.88)) and (FT3 β = −0.11 (−2.14, 1.96)). Lastly, a study by Sun et al. (2019) revealed that for each unit increase in the ln-transformed urinary As levels would lead to a 0.015 reduction in serum ln-FT3. Additionally, in an experimental animal study using rats, the serum levels of thyroid hormones in the treatment group exposed to As showed a significant decrease in comparison with control groups (p ≤ 0.05) (Maleki et al., 2019).

Cadmium

Cd was the most studied heavy metal in this review in which 86% (n = 25) of studies in this review assessed the association between Cd and thyroid hormones. Nine studies did not find any significant association between Cd and thyroid hormone disturbances (Al-Bazi et al., 2021; Campos et al., 2021; Castiello et al., 2020; Guo et al., 2018; Meeker et al., 2009; Nascimento et al., 2018; Sun et al., 2019; Xu et al., 2019, 2014). However, a significant association was proven in 12 human studies and four animal studies. Among the human studies, a significant association with TSH was observed in a study by Christensen (2013) in which Cd levels in both blood (adjusted β = −0.074 (0.028)) and urine ((adjusted β = −0.119 (0.055)) samples were associated with decreased TSH. This finding was further strengthened by a study in Greece (Margetaki et al., 2021) that found women with high (3rd tertile) concentrations of urinary Cd recorded 13.3% (95% CI [2.0–23.2%]) lower TSH compared to women with low concentrations (2nd and 1st tertiles). Another study in Poland (Jurdziak et al., 2018) also demonstrated a higher Cd level (aOR = 1.532; p = 0.027) as an independent risk factor of abnormal TSH levels among individuals in the occupationally exposed group. In addition, a study among pregnant women in China also showed a significant positive association between maternal exposure to Cd in the first trimester and neonatal TSH levels (p = 0.04) (Wang et al., 2020).

For T3 and T4 thyroid hormones, Cd in urine was associated with an increased T3 level (adjusted β = 0.033 (0.008)) and FT3 (adjusted β = 0.012 (0.004)) (Christensen, 2013). Another study in Korea found that urinary Cd was positively associated with total T3 in both male and female populations (β = 0.033, p < 0.001 in males and β = 0.032, p < 0.001 in females) (Kim et al., 2021). Furthermore, a positive association between Cd exposure and T4 (β = 0.02 (0.001, 0.03)) was also reported in a study in China (Chen et al., 2013). A similar positive correlation was also observed between Cd with serum-free T4 and T3 levels ((r = 0.167, p < 0.001 and r = 0.159, p < 0.001), respectively (Akgöl et al., 2017). This finding was further strengthened by a study in which blood Cd was positively associated with free T4 after adjustment for potential confounders (Luo & Hendryx, 2014). Additionally, a study in the USA also found that urinary Cd (OR: 2.05, 95% CI [1.03–4.06) was significantly associated with increased odds of thyroid dysfunctions (hypothyroidism and hyperthyroidism) (Liao & Zhang, 2019). Last but not least, two studies revealed a significant association between thyroid autoantibodies and Cd exposure (Chen et al., 2022; Nie et al., 2017).

With regard to experimental animal studies, three studies also show a significant association between Cd and the levels of thyroid hormones. A study by Luca et al. (2017) found a significant increase in the histological features of transformation in the thyroid follicular cells of rats treated with Cd compared with those in the control group. Another rat study in Iran found decreased serum levels of thyroid and parathyroid hormones in the treatment group in comparison with control groups (p ≤ 0.05) (Maleki et al., 2019). Lastly, an experimental study on rabbits by Khan et al. (2019) found that the level of serum T3 in the Cd exposed group (0.4 ± 0.0 ng/ml) recorded the highest significant decrease while the level of serum T4 (26.3 ± 1.6 ng/ml) also showed a comparable significant decrease. Moreover, the level of serum TSH in the Cd-exposed group (0.17 ± 0.01 nmol/l) showed the highest significant decrease. In contrast, only one experimental animal study showed a nonsignificant effect of Cd on the thyroid (Yu et al., 2018).

Mercury

The significant association between Hg and thyroid hormones was detected in seven human studies and one animal study in this review. In contrast, five studies showed a nonsignificant association between Hg and thyroid hormones (Campos et al., 2021; Guo et al., 2018; Meeker et al., 2009; Nascimento et al., 2018; Wang et al., 2020). A study in the USA found that Hg in blood (adjusted β = −0.013 (0.005)) and urine (adjusted β = −0.027 (0.012)) samples were associated with decreased total and free T3 and T4 levels (Christensen, 2013). Similarly, in another study, an inverse association was observed between Hg exposure and T4 hormone (β = –0.02 (–0.02, –0.01)) (Chen et al., 2013). This study was further strengthened by a study in Ghana in which blood Hg showed a negative correlation with T3 ((r = −0.29, p < 0.0001) and T4 (r = −0.69, p < 0.0001)). In contrast, urinary Hg was negatively associated with total T3 (β = −0.032, p < 0.001) and positively associated with total T4 among females only (β = 0.031, p = 0.016) in a Korean study (Kim et al., 2021). Another study also found a significant inverse association between Hg and TSH ((−4; 95% CI −8; −1 for each 50% increase in Hg)) (Castiello et al., 2020).

Among experimental animal studies, one rat study showed the effect of Hg on the level of thyroid hormone. Serum T3 in the Hg exposed group (0.4 ± 0.0 ng/ml) showed a higher significant difference compared to the control group, whereas the level of serum T4 in the Hg exposed group (21.3 ± 1.1 ng/ml) only showed a comparable significant decrease. The level of serum TSH in Hg exposed group (0.19 ± 0.01 nmol/l) showed the highest significant decrease difference (Khan et al., 2019). Similarly, an experimental rats study by Maleki et al. (2019) also found that serum levels of thyroid hormones in the Hg treatment group showed a significant decrease in comparison with control groups (p ≤ 0.05).

Lead

As for lead (Pb), a total of 12 studies in this review (10 human studies and two animal studies) observed the association with thyroid functions. However, 11 studies showed nonsignificant associations between Hg and thyroid hormones (Campos et al., 2021; Castiello et al., 2020; Chen et al., 2013; Guo et al., 2018; Jurdziak et al., 2018; Kim et al., 2021; Li et al., 2017; Liao & Zhang, 2019; Xu et al., 2019, 2014; Yalçin et al., 2022). A study by Meeker et al. (2009) found that Pb was associated with a reduction in TSH (β = −0.19 (−0.38, 0.004)). Furthermore, another study observed that the natural log(ln) Pb was positively related to the thyroid peroxidase antibody (B¼ = 0.062, p < 0.05) and the TSH (B¼ = 0.047, p < 0.01) levels in women (Nie et al., 2017). Similarly, another study found that blood levels of Pb (r = 0.376, p < 0.05) were positively correlated with TSH concentrations and negatively associated with FT4 levels in the low exposure period (Nascimento et al., 2018).

Furthermore, a study in the USA observed that blood Pb level was positively associated with free T3 but not with any other thyroid hormones after adjustment for the same confounding variables (Luo & Hendryx, 2014). This finding was echoed by a study in Greece that found women with high urinary Pb had 4% (95% CI [0.2–8.0%]) higher fT3 levels compared to women with low exposure (Margetaki et al., 2021). Another study found that Pb in both blood (adjusted β = −0.028 (0.013)) and urine (adjusted β = −0.034 (0.013)) samples were also associated with decreased T4 levels (Christensen, 2013). In contrast, another study in China found that urinary Pb concentration exhibited an inverse association with the FT3 or FT3/FT4 ratio. For each unit increase in the ln-transformed urinary Pb level, there was a reduction of 0.015 and 0.011 in serum ln-FT3 (Sun et al., 2019). A study among occupational exposed workers to Pb also found that mean blood Pb levels (16.5 ± 1.74 μg/dl) were significantly higher compared to those in the control group ((12.8 ± 1.16 μg/dl, p < 0.001)). The exposed group also had significantly increased free FT3 and FT4 levels (p < 0.0001) and significantly decreased TSH level with mean (1.77 ± 0.44 μIU/ml) (p < 0.0001) (Fahim et al., 2020). Finally, a study in China found that participants with the fourth quartile of Pb were positively associated with a higher level of thyroid peroxidase antibody (OR 1.637, p = 0.006), antithyroid antibody (OR 1.435, p = 0.025), hypothyroid status (OR 1.467, p = 0.013), and TSH levels (β = 0.092, (p = 0.021)) (Chen et al., 2022).

In contrast, an animal study revealed that serum thyroid levels in the Pb treatment group showed a significant decrease in comparison with control groups (p ≤ 0.05). In addition, histological investigations demonstrated relative changes in tissue and functional structures of important thyroid gland tissues (Maleki et al., 2019). Another experimental rat study also shows that Pb increased serum levels of T4 by 25.7% (p < 0.05) in the male group treated with 25 mg/kg lead acetate and both female groups treated with 10 and 25 mg/kg of lead acetate by 58.8% and 52% respectively (p < 0.05). Comparatively, T3 serum levels were reduced by 50.9% in the male group treated with 10 mg/kg of lead acetate and 20.3% in the female group treated with 25 mg/kg. However, when compared to the respective control groups, TSH serum levels did not show statistically significant changes (de Lima Junior et al., 2021).

Discussion

Five heavy metals were evaluated in the studies included in this review, namely Cr, Ar, Cd, Hg, and Pb. Several significant associations between heavy metals and noncancerous thyroid illness from this review and other studies were discussed. The mechanism of action involved was also elaborated and any nonsignificant association was also compared with other studies. Notably, most of the nonsignificant associations of the studies were due to heterogeneity in the sample population, sample selection, human biomarker sample type, and geographical differences. In the section below, we discuss the key findings for each metal, contextualize the results with broader literature, and propose directions for future research. To facilitate a more structured discussion, the association between different types of toxic heavy metal elements with thyroid function will be outlined in the order of the atomic number of each heavy metal in the periodic table.

Chromium (atomic number 24)

The association between Cr and noncancerous thyroid disease can be explained by the mechanism proposed by the animal study conducted by Mohamed & Abd El-Twab (2016) in which Cr causes thyroid dysfunction by disrupting inflammatory cytokines and antioxidant levels, subsequently affecting thyroid follicles and thyroid homeostasis. Another animal study by Mahmood, Qureshi & Iqbal (2010), also found hyperplasia on the portions of the Cr-treated thyroid; some follicles were fused, while the others were haphazardly grouped, crumpled, or presented with chaotic features. In addition, the gaps between follicles also grew. The results of the morphometrical study revealed that while follicular size dramatically decreased, the number of follicles grew significantly. Furthermore, regressed nuclei had unusual nuclear forms, including bony, oval, and round morphologies. Both the basal lamina side and the apical membrane were disrupted. Serum concentrations of TSH increased (p < 0.01), but serum levels of FT3 and FT4 dropped (p < 0.01 and p < 0.001) respectively (Mahmood, Qureshi & Iqbal, 2010).

Overall, our review found a significant association between Cr and noncancerous thyroid disease, suggesting that Cr exposure may influence thyroid function. However, several studies reported nonsignificant associations in our review, highlighting variability in findings. Furthermore, Liu et al. (2021) reported a significant inverse correlation between Cr exposure and the incidence of thyroid goiter in a single-element model. This discrepancy underscores the complexity of the relationship between Cr and thyroid health, which may be further influenced by varying study designs, populations, and exposure assessment methods used in different studies. While this study was not included in our review, its findings add valuable insight into potential geographical and methodological differences affecting the overall outcome. Future research should aim to clarify these discrepancies by standardizing exposure assessments and focusing on populations at higher risk of Cr exposure.

Arsenic (atomic number 33)

The association between As and noncancerous thyroid disease can be explained by the mechanism proposed by a study in China in which rats exposed to NaAsO2 had increased serum TSH and decreased serum T3 and T4 levels, histological alterations in the thyroid, raised ratio of Bax/Bcl-2, and buildup of As in thyroid tissues (Fan et al., 2023). Besides that, another animal study also found that even at extremely low concentrations, As significantly impacted a thyroid receptor-dependent developmental process in an animal model. Its impact also extends to thyroid receptor-mediated gene regulation (Davey et al., 2008). Apart from that, human As thyrotoxicotic research also showed that increased As exposure dramatically raised study participants’ levels of thyroglobulin and TSH while lowering their levels of free T4 and T3 that are freely circulated in the bloodstream unbound by albumin, transthyretin, and thyroid-binding globulin (Ciarrocca et al., 2012). Lastly, As may also inhibit thyroid peroxidase regulation and activity in a dose-dependent relationship (Palazzolo & Ely, 2015; Palazzolo & Jansen, 2008).

In this review, several studies reported a significant association between As and noncancerous thyroid disease. However, some other studies did not find any significant association. This inconclusive finding underscores the need for longitudinal studies examining chronic low-dose As exposure to elucidate its long-term thyroid effects, particularly in regions with As-contaminated water supplies.

Cadmium (atomic number 48)

Cd was the most extensively studied metal in this review. The association between Cd and noncancerous thyroid disease can be explained by a mechanism proposed by Benvenga et al. (2020), Jancic & Stosic (2014), Pavia Júnior et al. (1997), and Sola et al. (2022) whereby numerous histological and metabolic alterations in the thyroid gland can be brought on by long-term exposure to Cd. In most studies, Cd has been found to directly reduce or increase the activity of certain enzymes, interfere with the proper synthesis of membrane and secretory proteins, reduce the number of antioxidants inside cells, and suppress the production of antioxidant enzymes (Chen et al., 2023; Jancic & Stosic, 2014; Sola et al., 2023; Unsal et al., 2020). Furthermore, studies have also demonstrated that Cd can activate or stimulate many factors that increase cell proliferation and decrease normal apoptotic activity in the thyroid (Buha et al., 2018; Shin et al., 2003; Wade et al., 2002). A meta-analysis by Chung & Chang (2023) revealed that Cd exposure was positively associated with elevated levels of T3, hence suggesting a thyroid-disrupting effect that could contribute to thyroid dysfunction without necessarily leading to cancer. Moreover, studies have indicated that Cd can also interfere with thyroid hormone homeostasis and immune function, as evidenced by its effect on TgAb (Alerte et al., 2022).

In addition, Shao et al. (2023) found that higher Cd levels were associated with lower TSH levels and an increased risk of thyroid dysfunction. However, the effect varies by sex; men showed more significant changes in TSH levels in response to Cd than women (Shao et al., 2023). Furthermore, animal research suggests that Cd exposure may lead to structural damage and inflammation in thyroid tissues by promoting oxidative stress and apoptosis, all of which could further disrupt thyroid functions (Chen et al., 2023). However, other studies in this review discovered no significant correlation between Cr and thyroid conditions. These discrepancies may stem from underlying differences in biomarker selection, exposure assessment, or population demographics. Therefore, future research should investigate the cumulative effects of Cd and its interactions with other environmental stressors, focusing on vulnerable groups such as pregnant women and children.

Mercury (atomic number 80)

Hg buildup in the thyroid can cause particular cytotoxicity in thyroid tissues, leading to hormonal fluctuations that affect the up or down-regulation of thyroid metabolism-related enzymes (Tan, Meiller & Mahaffey, 2009). Moreover, exposure to Hg may lower deiodinase activity and this in turn may result in a reduction in T3. However, there is little knowledge of the mechanism of action and adverse effects of Hg in humans. Furthermore, numerous animal studies revealed that Hg can have different effects on the thyroid, depending on the type of experimental animal used, the chemical forms of Hg, as well as the difference in the dose, method, and duration of administration (Zhu et al., 2000).

In our review, several studies show a significant correlation between Hg levels and noncancerous thyroid illness. However, a nonsignificant association between Hg and noncancerous thyroid disease was also reported in some studies. To address these gaps, future studies should prioritize the understanding of the differential effects of Hg across populations and chemical forms. In particular, research should focus on populations with high occupational or environmental exposure to Hg.

Lead (atomic number 82)

Animal studies by Der et al. (1977) and Lu et al. (2022) attempted to explain the link between Pb and noncancerous thyroid illness. Pb was found to modify the tissue structure, antioxidant capacity, hormone release, and gene levels of the thyroid, leading to a negative impact on the endocrine system. Another study found that Pb induction of sex-biased thyroid autoimmunity could be due to the correlation between Pb exposure with greater TSH in women (Nie et al., 2017). Furthermore, animal studies using Wistar rats also suggested thyroid gland dysfunction due to subacute Pb exposure. The study found a decrease in colloid eosinophilia and vascular congestion, aberrant thyroid parenchyma follicles of varying sizes, epithelial stratification and vacuolization of follicular cells, and morphometric changes in the thyroid gland. Additionally, there was an increase in the deposition of collagen. However, no variations were seen in the expression of hepatic D1 mRNA, nicotinamide adenine dinucleotide phosphate (NADPH) oxidases production of H2O2, thyroid peroxidase (TPO) activity, or protein expression. However, there was a significant decrease in thyroid sodium/iodide symporter (NIS) protein expression among individuals undergoing Pb treatment (de Lima Junior et al., 2021). Mixed findings of significant and nonsignificant associations between Pb and noncancerous thyroid illness have also emerged from our review. Therefore, further research is imperative to explore the interaction between Pb with other environmental toxicants and its long-term health impacts.

Limitations

In view of the significant differences across the studies included in this scoping review in terms of population, exposure assessment techniques, and outcome measures, it is challenging to achieve conclusive findings on the correlation between hazardous heavy metals and thyroid disorders. Additionally, we did not include a quantitative synthesis of the data in this scoping review, hence making it more difficult to gauge the degree of correlation between toxic heavy metals and non-malignant thyroid disorders. Lastly, most of the results in this scoping review can only be considered as preliminary and hypothesis-generating rather than definitive conclusions. To establish causal linkages, more research is typically required, especially longitudinal primary studies and systematic reviews.

Conclusion

A heterogeneous and diverse sample population in the studies included in this review has led to mixed findings of both significant and nonsignificant associations between the main toxic heavy metals and thyroid illness. Specifically, the following conclusions can be drawn from the review: 1) Cr: While a small number of studies indicate a significant association with thyroid dysfunction, the majority of the studies report nonsignificant findings. Future research should explore the impact of Cr exposure using longitudinal designs and refined exposure metrics to clarify these inconsistencies. 2) As: Evidence indicates significant associations between As exposure and thyroid hormone alterations, particularly in populations with high exposure. Prospective cohort studies are needed to examine the long-term thyroid health effects of As, especially in areas with As-contaminated water supplies. 3) Cd: As the most studied metal in the studies, Cd shows a strong association with thyroid hormone disruptions. More studies are needed to elucidate the underlying mechanisms to implement population-level interventions that can mitigate Cd exposure. 4) Hg: Mixed findings on the effects of Hg on the thyroid highlight the need for research into its differential effects based on gender and other demographic variables. Studies should assess thyroid-specific risks among populations with occupational exposure to Hg. 5) Pb: While Pb has shown consistent associations with thyroid autoimmunity and dysfunction, further research should investigate its cumulative impact on thyroid health over the lifespan, especially in children and pregnant women. In short, this review underscores the need for a unified and interdisciplinary approach to addressing the impact of environmental heavy metals on thyroid health. Bridging the fields of environmental science, clinical endocrinology, and public health can provide a comprehensive understanding of these associations and drive innovative solutions to protect vulnerable populations. More importantly, future research should prioritize longitudinal studies and controlled clinical trials to ascertain the causative mechanisms and long-term impacts of heavy metal exposure on thyroid health.

Supplemental Information

Supplemental Information 1 Study Characteristics.

Supplemental Information 2 Association between heavy metals (Cr, As, Cd, Hg, Pb) and noncancerous thyroid illness.

*Statistical analysis used. **Tested heavy metals but shows nonsignificant association with noncancerous thyroid illness in the study.

Supplemental Information 3 PRISMA Checklist.

Supplemental Information 4 PRISMA 2020 flow diagram.

The Department of Community Health at UKM and the Faculty of Medicine were helpful in the conduct of this study, which the authors gratefully acknowledge.

Additional Information and Declarations

Competing Interests

The authors declare that they have no competing interests.

Author Contributions

Muhammad Ridzwan Rafi’i conceived and designed the experiments, performed the experiments, analyzed the data, prepared figures and/or tables, authored or reviewed drafts of the article, and approved the final draft.

Mohd Hasni Ja’afar conceived and designed the experiments, performed the experiments, analyzed the data, authored or reviewed drafts of the article, and approved the final draft.

Azmawati Mohammed Nawi conceived and designed the experiments, performed the experiments, analyzed the data, authored or reviewed drafts of the article, and approved the final draft.

Shahrul Azhar Md Hanif conceived and designed the experiments, performed the experiments, analyzed the data, prepared figures and/or tables, and approved the final draft.

Siti Najiha Md Asari conceived and designed the experiments, performed the experiments, analyzed the data, prepared figures and/or tables, and approved the final draft.

Data Availability

The following information was supplied regarding data availability:

This is a literature review.

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
