# Peer review of "Association between toxic heavy metals and noncancerous thyroid disease: a scoping review"

_PeerJ, doi:10.7717/peerj.18962_

## Round 0.1 · original submission · Major Revisions

Please, carefully follow the referees' suggestions. The article needs a deep revision

·

Basic reporting

Dear Editor, In my opinion, the manuscript entitled " Association between toxic heavy metals and noncancerous thyroid disease: A scoping review ((#105627)) The information that has been provided is of good/new value to readers. However, major revisions are necessary. Therefore, I would like to make some suggestions.
1. The review may not receive broad and interdisciplinary attention and should be developed.
2. Many expressions appear to be not fluent, which significantly decrease the readability.
In the introduction, it can be enriched with a few sentences about heavy metals and noncancerous thyroid disease.
3. In the conclusion section in the abstract section, in vitro studies or further studies. The authors gave clinical trials as examples. Can in vitro / in vivo be recommended when there are clinical findings? A more scientifically appropriate sentence should be written.
4. A more appropriate sentence can be made in the abstract section. Heavy metals can only have an effect on tissues or organs instead of environmental pollutants. This study is not about the environment. Heavy metal toxicity can be addressed.
5. There are no keywords, if they are available in journal format, they should be added.
6. Unfortunately, there is a lot of repetition of sentences in the Discussion part. The Results and Discussion sections are similar to each other. Therefore, the Discussion section should be discussed more clearly and clearly.

Experimental design

No comment

Validity of the findings

The impact and novelty of this study may be partially effective, but I leave it to the Editor to evaluate whether the existing results/findings and discussion are sufficient after being enriched.

Additional comments

Dear Editor, The study partially contributes to the literature, but the study should be written in a more scientific, professional and understandable way.

·

Basic reporting

The topic is important because of the increased exposure of the world's population to toxic substances and environmental pollution.

Experimental design

The methodology and presentation of the search results are adequate.

Validity of the findings

The novelty of the information presented is adequate, the tables are clear and although the heterogeneity and design of the selected and analyzed studies does not allow us to conclude about a real association of the effects of these metals on thyroid function.

Additional comments

The syntax in English of the manuscript should be reviewed.

·

Basic reporting

The manuscript contains a very broad analysis - searching over a wide date range, covering both blood and urine samples, from both animals and humans, for up to 5 elements.
There is some ambiguity regarding thyroid tissues - when describing the methodology, the authors do not mention whether they included articles analyzing thyroid tissue samples or not. However, in the results section there are fragments regarding tissues, for example: “A study by (Luca et al. 2017) found a significant increase in histological features of transformation was observed in thyroid follicular cells of rats treated with Cd compared with those of the control group.” or „In addition, histological investigations demonstrated relative changes in tissue and 318 functional structures of important thyroid gland tissues (Maleki et al. 2019).”
It is true that the references to tissues do not appear in the context of the analysis of how many heavy metals were in this tissue, but about the histology of the tissues. I believe that it would be better to organize this in the manuscript, in one of two ways:
1. Include analyses of the content of heavy metals in intraoperative thyroid biopsies and clearly indicate this in the manuscript; or
2. Remove the inclusions about tissues, at least in the Results section.

The Introduction adequately introduce the subject and make it clear what the motivation of the authors is.

Experimental design

The methodology is unqualified except for the tissue aspect, which I described in 1 point.

The sources are not always adequately cited. For example, in the passage with lines 189-195: Why is Li et al cited but not cited for the data in the previous sentence? I recommend that you study the entire manuscript to verify that a source is provided for each data item. I also suggest that you organize the citation of data from sources, e.g. "(B 0.092, p=0.021)" line 311 or "β = (0.013)" line 300 – shouldn't it be B or β, equals sign, bracket?
Another type of inconsistency in citing results is when only half of the data from the cited publication is given, and some are omitted for some unknown reason, e.g. in lines 199-202 r and p are quoted for the Cr-TSH correlation, but these values are not quoted for the Cr-fT4 correlation.

The review organized logically into coherent paragraphs, although the content of the results and discussions is largely repeated in both parts, or one even gets the impression that the "discussion" is an extension of the "results", i.e. a continuation of the results. In the discussion, additional publications were added, which were apparently found in a different way than described in the methodology. For example, for Chromium - in the Result section, 7 publications are described. In the discussion, this data is repeated plus data from other publications is added.
I quote: Results “For the nonsignificant association between Cr and thyroid hormone levels, there were 5 studies showed the findings in this review (Al-Bazi et al.2021; Guo et al. 2018; Meeker et al. 2009; Sun et al. 2019; Xu et al. 2019)”
I quote: Dissusion “In contrast, there are 5 studies in this review found a nonsignificant association between Cr and noncancerous thyroid disease. These findings are also supported by other findings of a study in China that found the incidence of thyroid goiter was significantly inversely correlated with Cr according to single-element models (Liu et al. 2021).”(I leave aside the fact that the sentence is illogical to me, because “These findings are NOT supported by other findings of a study in China ( …)” ).
--> Why is Liu et al not in Result, it only appears in Discussion? I suspect that the methodological search did not find this work. So why is it added to the discussion, on the basis of "adding results"?
This is just an example. In general, I think the discussion should be reworked, removing unnecessary fragments and making it more reflective, and its goal is primarily to outline future research possibilities and indicate how the results so far may affect the further development of knowledge in a given area.

Validity of the findings

The conclusions are too general. No specific conclusion about a specific element or any specific area that should be studied is indicated. In my opinion, more specific conclusions should be proposed.

---

## Round 0.2 · Minor Revisions

Please, highlight the remaining issues within the manuscript.

·

Basic reporting

Dear Editor, the requested revisions have been made, but I have some small suggestions to strengthen the article and make it more cited. Especially in the Introduction and the definition of the Elements, I still have concerns about the references and strengthening the article, so I will suggest a few article references (if possible, from the last 5 years). I think that the authors will improve the article after reading it quickly and citing it.

1.https://doi.org/10.1016/j.chemosphere.2024.142691
2.doi: 10.34172/apb.2020.023
3.https://doi.org/10.1016/j.molstruc.2024.139338
4.10.15171/apb.2018.043
5.https://doi.org/10.1007/s11356-022-22705-6

Example; Toxic heavy metals such as Chromium (Cr), Arsenic (As), Cadmium (Cd), Mercury (Hg), and Lead (Pb) have been identified as priority pollutants due to their high degree of toxicity and widespread presence in the environment (Balali-Mood et al. 2021; XXX el at., 2024; XXX)

Experimental design

Article content is within the Aims and Scope of the journal and article type.

Validity of the findings

Conclusions are well stated, linked to original research question & limited to supporting results.

·

Basic reporting

no comment

Experimental design

no comment

Validity of the findings

no comment

Additional comments

Interesting topic developed and updated on pollution in thyroid diseases, although the literature does not support obtaining robust conclusions, there seems to be a tendency for an association between thyroid diseases and heavy metals.

·

Basic reporting

No comment.

Experimental design

No comment.

Validity of the findings

No comment.

Additional comments

Thank you very much for your feedback. The work has been significantly improved, although some comments still remain:
1. In response to my first comment and in the manuscript in line 182 the same mistake was made by writing "biomarkers such as blood or urine". Blood and urine are not markers, but biological materials in which biomarkers can be sought, although one must be aware that elements will never be markers. Please correct.
2. Unfortunately, the authors in each case provided misleading references to places where they corrected fragments. For example, it was written in the first comment "Please kindly refer to the line 146-150 and the line 156-159". There is nothing in these lines concerning the case. Probably it refers to lines 182-186 and 193-195. The same is true for each reference to a fragment. This makes it very difficult to revise the manuscript again.
3. In general, I still see inconsistency in reporting results - although "p" is sometimes a capital letter, sometimes a lower case one, sometimes "95% CI" sometimes just "CI" and other incomprehensible entries, e.g. Line 380: "58.8% e 52%", - what does "e" mean?) - here I ask the authors to review the manuscript again with a person familiar with statistics to standardize the entries.
However, these are such minor issues that I decide to give the opinion "accept with minor revisions"

---

## Round 0.3 · accepted · Accept

I have assessed the revision myself, and that I am happy with the current version.